# Investigating Self-supervised Pre-training for End-to-end Speech Translation

Ha Nguyen [1 2]   Fethi Bougares [3]   Natalia Tomashenko [2]   Yannick Estève [2]   Laurent Besacier [1]

## Abstract

Self-supervised learning from raw speech has been proven beneficial to improve automatic speech recognition (ASR). We investigate here its impact on end-to-end automatic speech translation (AST) performance. We use a contrastive predictive coding (CPC) model pre-trained from unlabeled speech as a feature extractor for a downstream AST task. We show that self-supervised pre-training is particularly efficient in low resource settings and that fine-tuning CPC models on the AST training data further improves performance. Even in higher resource settings, ensembling AST models trained with filter-bank and CPC representations leads to near state-of-the-art models without using any ASR pre-training. This might be particularly beneficial when one needs to develop a system that translates from speech in a language with poorly standardized orthography or even from speech in an unwritten language.

## 1. Introduction

Self-supervised learning using huge unlabeled data has been explored with very promising results for image processing (Chen et al., 2020) and natural language processing (Devlin et al., 2018). Recent works investigated self-supervised representation learning from speech (Baevski et al., 2019; Kawakami et al., 2020; Chung & Glass, 2019). They were successful to improve performance on downstream tasks such as speech recognition. These recent works suggest that it is possible to reduce dependence on labeled data for building speech systems through acoustic representation learning. We investigate the possibility to leverage unlabeled speech for end-to-end automatic speech translation (AST). We focus on scenarios where (a) recordings in source language are

not transcribed[1] (no ASR pre-training is possible), (b) only a small-medium amount of training data (speech aligned to translations) is available, (c) a larger amount of unlabeled speech can be used. This scenario is typical of situations when one builds a system that translates from speech in a language with poorly standardized orthography or even from an unwritten language.

In summary, our contributions are: (1) we propose an in-depth study on the impact of self-supervised pre-training for AST, (2) we show that fine-tuning pre-trained representations on the AST training data is beneficial and that self-supervised pre-training is particularly efficient in low resource settings, (3) even in high resource settings, ensembling models trained with filter-bank and self-supervised representations leads to near state-of-the-art models without using ASR pre-training, (4) we analyze the representations learnt and show that they allow to better discriminate phones, better align source and target sequences, and are more robust to speaker variability.

## 2. Related Works

### 2.1. Self-supervised learning from speech

Self-supervised learning from speech consists in resolving pseudo-tasks not requiring human annotations as a pre-training to the real tasks to solve. These pseudo-tasks target predicting next samples or solving ordering problems. Autoregressive predictive coding (APC) (Chung et al., 2019; Chung & Glass, 2020) considers the sequential structure of speech and predicts information about a future frame. An easier learning objective is introduced in Contrastive Predictive Coding (CPC) which consists in distinguishing a true future audio frame from negatives (Baevski et al., 2019; Schneider et al., 2019; Kahn et al., 2019). (Chung & Glass, 2019) shows that such representations are useful to improve several speech tasks while (Kawakami et al., 2020) extends those works by looking at the representations' robustness to domain and language shifts. In the same vein, (Rivière et al., 2020) compares self-supervised and supervised pre-training for ASR and shows that CPC pre-training extracts features that transfer well to other languages, being on par or even outperforming supervised pre-training. Another promising

---

[1]LIG - Université Grenoble Alpes, France [2]LIA - Avignon Université, France [3]LIUM - Le Mans Université, France. Correspondence to: Ha Nguyen <manh-ha.nguyen@univ-grenoble-alpes.fr>.

*Published at the workshop on Self-supervision in Audio and Speech at the 37th International Conference on Machine Learning*, Vienna, Austria. Copyright 2020 by the author(s).

---

[1]Transcription not available or language poorly written

way is to use speech enhancement as a task for feature representation learning (Ravanelli et al., 2020; Engel et al., 2020). Finally, several self-supervised tasks can be jointly tackled to discover better speech representations (Pascual et al., 2019).

## 2.2. End-to-end Automatic Speech Translation

Previous automatic speech-to-text translation (AST) systems operate in two steps: source language automatic speech recognition (ASR) and source-to-target text machine translation (MT). However, recent works have attempted to build end-to-end AST without using source language transcription during learning or decoding (Bérard et al., 2016; Weiss et al., 2017) or using it at training time only (Bérard et al., 2018). Recently several extensions of these pioneering works were introduced: low resource AST (Bansal et al., 2018), unsupervised AST (Chung et al., 2018), end-to-end speech-to-speech translation (*Translatotron*) (Jia et al., 2019), multilingual AST (Di Gangi et al., 2019). Improvements of end-to-end AST were also proposed using weakly supervised data (Jia et al., 2018) or adding a second attention mechanism (Sperber et al., 2019). While supervised pre-training for AST was investigated (see for instance (Bérard et al., 2018)), we are aware of a single research group (Chung & Glass, 2019; 2020) that investigated self-supervised pre-training for AST. However their experiments were done in a high resource setting and AST (for which only marginal gains were displayed) was solely investigated among other tasks, without an in-depth analysis of the representations learnt.

## 3. Self-supervised Pre-training from Speech

### 3.1. Contrastive predictive coding model

We use the self-supervised pre-training model introduced in (Schneider et al., 2019) (*wav2vec*) which is based on contrastive predictive coding. The model uses (1) an encoder network that converts the audio signal in a latent representation (from raw speech samples $x$ into a feature representation $z$), and (2) a context network that aggregates multiple time steps to build contextualized representations (from a sequence $z_{i-v}, ..., z_i$ into a context vector $c_i$).[2] The full model (encoder+context) is trained end-to-end to distinguish a sample $z_{i+k}$ that is k steps in the future from negative samples $\tilde{z}$ uniformly chosen from the same audio sequence. A contrastive loss is minimized for each step $k = 1, ..., K$ and the overall loss is summed over different step sizes (more details in (Schneider et al., 2019)).

*Table 1.* Statistics of different How2 data partitions

| Partition | #segments | #hours | #src_w | #tgt_w |
|---|---|---|---|---|
| 10% | 17,751 | 28 | 313K | 295K |
| 20% | 35,858 | 56 | 626K | 591K |
| 30% | 53,698 | 84 | 887K | 940K |
| 60% | 107,676 | 169 | 1778K | 1883K |
| full | 179,438 | 281 | 2963K | 3139K |

### 3.2. Pre-trained models for English

We use an off-the-shelf model provided for English.[3] It is trained on *Librispeech* corpus (Panayotov et al., 2015). We also investigate if fine-tuning the model on our task specific data is beneficial. For this, we fine-tune *wav2vec* on the full speech corpora used for our AST experiments (see next section). It is important to note that no transcripts nor translations are needed for this step which requires only raw speech. After fine-tuning *wav2vec*, we input the representations produced by the context network $c_i$ to the AST encoder instead of filter-bank features (see Figure 1).

## 4. End-to-end Speech Translation Experiments

### 4.1. Experimental setup

#### 4.1.1. DATA

How2 corpus (Sanabria et al., 2018) is used for our main experiments. This corpus contains about 297.6 hours of speech, which is transcribed and translated into 3.3 million of English words and 3.1 million of Portuguese words respectively.[4] From this version of data, we first filter out too long sentences (sentences longer than 30 seconds or 400 characters). Then, in order to simulate lower resource scenarios, we randomly split the corpus into four sub-corpora of roughly 10%, 20%, 30%, and 60% of the filtered full corpus. Our splits guarantee that smaller partitions are fully included in the bigger ones. The statistics of all the partitions and the filtered version of full corpora can be found in Table 1.

#### 4.1.2. SPEECH FEATURES AND DATA AUGMENTATION

As shown in Figure 1, we extract either *wav2vec* features or filter-bank+pitch features (later denoted as *fbanks*) from speech input.[5] Depending on the experiments, mean and

---

[2]Practically, each $z_i$ encodes $30ms$ of speech every $10ms$. As for $c_i$, the total receptive field of the context network is $210ms$.

[3]https://github.com/pytorch/fairseq/blob/master/examples/wav2vec/

[4]As shown by (Nguyen et al., 2019), How2 is sensitive to the downloading moment. Our version was downloaded in July, 2019.

[5]Our preliminary experiments on How2 10% with MFCC features which lead to similar performance as filter-bank are not

variance normalization (*MVN*) is optionally applied to the generated features. For *wav2vec* feature extraction, we either use an off-the-shelf model trained on LibriSpeech (Panayotov et al., 2015) or a model fine-tuned on How2 training set. *MVN* parameters are estimated on the speech translation training set and then applied to all train/dev/test sets. Overall, we have 4 different self-supervised representations named *wav2vec*, *wav2vec + norm*, *wav2vec + FT* (fined-tuned *wav2vec*) and *wav2vec + FT + norm*. All those *wav2vec* features are of dimension 512. We compare the above representations to conventional *filter-bank* features. Similar to (Nguyen et al., 2019), we extract 80-dimensional Mel filter-bank features, concatenated with 3-dimensional pitch features from windows of $25ms$, and a frame shift of $10ms$. *MVN* is used in the same manner as for *wav2vec* features. This gives us 2 additional speech representations named *fbanks* and *fbanks + norm* respectively (their dimension is 83).[6] Data augmentation through speed perturbation is also applied with factors of $0.9$, $1.0$, and $1.1$ to the training data. Our development set is made of $1,984$ sentences randomly excluded from the training set. How2 val set is used as our test data.

## 4.2. Speech-to-text translation model

### 4.2.1. ARCHITECTURE.

We use an attention-based encoder-decoder architecture, whose encoder is illustrated in Figure 1. The encoder is a stack of two VGG-like (Simonyan & Zisserman, 2015) CNN blocks followed by five 1024-dimensional BLSTM layers. Each VGG block contains two 2D-convolution layers just before a 2D-maxpooling layer, which aims to reduce both time ($T$) and frequency dimension ($D$) of the input speech features by a factor of 2. These two VGG blocks transform input speech features' shape from ($T \times D$) to ($T/4 \times D/4$). Bahdanau's attention mechanism (Bahdanau et al., 2015) is used in all our experiments. The decoder is a stack of two 1024-dimensional LSTM layers. As proven effective in (Nguyen et al., 2019), this model is consistently used for all the experiments with *fbanks* features presented throughout this paper. However *wav2vec* features have higher dimension (512) than *fbanks* (83). In order to compare both input representations with a similar parameter budget in the architecture (and also because training an architecture with input features of dimension 512 would be substantially more computationally expensive), we add a projection block at the bottom of the encoder.[7] This block (containing a linear

---

presented here.

[6]For the rest of the paper *fbanks* will actually mean filter-bank+pitch

[7]Our implementation of the *wav2vec* speech encoder, as well as the detailed recipes for our experiments can be found online: https://github.com/mhn226/espnet/tree/interspeech2020.

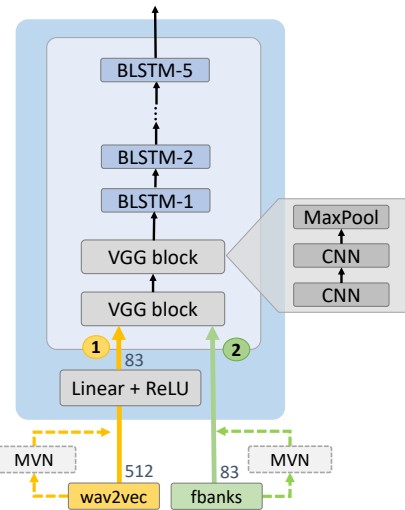

*Figure 1.* Architecture of the speech encoder: a stack of two VGG blocks followed by 5 BLSTM layers. We use as input (1) *wav2vec* features (that pass through an additional projection layer to reduce their dimension from 512 to 83), or (2) filter-bank+pitch features. The input features are optionally normalized (MVN).

layer followed by a ReLU) reduces the *wav2vec*'s feature size from 512 to 83 (see Figure 1).

### 4.2.2. HYPERPARAMETERS' DETAILS

Models are trained in maximum 20 epochs with early stopping after 3 epochs if the accuracy on the dev set does not improve. Adadelta is chosen as optimizer and dropout is set to $0.3$ on the encoder side. We decode all our models with beam size of 10.

## 4.3. Experimental results on How2

On each partition of How2 corpus, we train 6 models which take as input different speech representations presented in section 4.1.2, thus in total 30 models shown in Table 2. We evaluate on How2 val set, which contains $2,022$ segments (about 3.2 hours of speech), in the same conditions as (Nguyen et al., 2019). It is clear from the table that in low resource settings (28 and 56 hours), self-supervised representations (*wav2vec*) significantly outperform *fbanks*. Figure 2a confirms this and shows that models trained with *wav2vec* representations converge better and faster. The impact of normalization and fine-tuning is also notable from both Table 2 and Figure 2a. In very low resource settings (like 28 hours), fine-tuning *wav2vec* can greatly help, and with normalization, the performance further improves. In higher resource settings (169 and 281 hours of translated speech), differences between *wav2vec* and *fbanks* fade away (and so does the impact of fine-tuning and normalization). However, our ensembling experiments of lines 7 and 8 on

*Table 2.* Detokenized case-sensitive BLEU scores measured on How2 val set of different models trained on different partitions of How2 corpus (EN-PT) with different speech features. **FT** means fine-tuned and **norm** stands for MVN normalization.

| No. | Feature | 10% (28h) | 20% (56h) | 30% (84h) | 60% (169h) | 100% (281h) |
|---|---|---|---|---|---|---|
| 1 | wav2vec | 11.33 | 26.75 | 30.83 | 36.33 | 41.02 |
| 2 | wav2vec + FT | 12.52 | 27.30 | 32.11 | 37.78 | 42.32 |
| 3 | wav2vec + norm | 16.52 | 27.33 | 31.27 | 37.62 | 41.08 |
| 4 | wav2vec + FT + norm | 18.50 | 27.68 | 32.17 | 37.75 | 41.30 |
| 5 | fbanks | 1.03 | 18.61 | 27.32 | 37.23 | 41.63 |
| 6 | fbanks + norm | 2.11 | 24.58 | 30.21 | 37.56 | 42.51 |
| 7 | Ensemble [5, 6] | | 25.28 | 31.90 | 40.39 | 44.35 |
| 8 | Ensemble [4, 6] | | 29.87 | 34.67 | 41.22 | 45.02 |
| 9 | **Ensemble [1,2,3,4,5,6]** | | **31.88** | **36.80** | **42.62** | **46.16** |

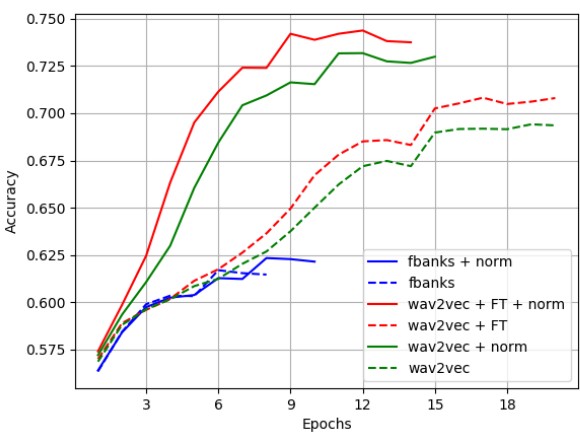

(a) How2 10% (28 hours)

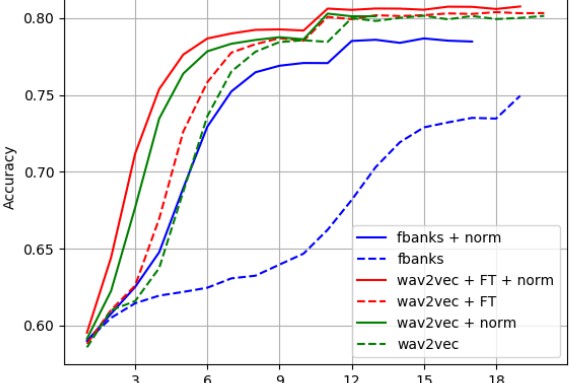

(b) How2 20% (56 hours)

*Figure 2.* Learning curves (accuracy) of models trained on different partitions of How2

100% of How2 show that it is beneficial to ensemble the best system (*fbanks+norm*, line 6) with a system trained with *wav2vec* (*wav2vec+FT+norm*, line 4) rather than a better model (*fbanks*, line 5) also based on *filter-bank* features, even though *wav2vec+FT+norm* underperforms *fbanks* on this partition. Ensembling all our models (line 9) leads to $BLEU > 30$ even in very low resource training conditions (56 hours). Finally, in order to compare ourselves with the state-of-the-art (Inaguma et al., 2020), we decode How2 dev5 (a.k.a How2 test), which consists of $2,305$ segments (about 3.7 hours of speech), using the ensemble of all our models trained on the full corpus (line 9). This gives us near state-of-the-art BLEU: we obtain $46.16$ on How2 val and $47.17$ on How2 dev5. This latter score on dev5 is to be compared with $48.04$ reported with an ensemble model in (Inaguma et al., 2020) where ASR and MT pre-training were used, as well as data augmentation with *SpecAugment*.

### 4.4. Validation on two other language pairs

To validate our results in low resource settings (56 hours), we train our models on two subsets of MuST-C (Di Gangi et al., 2019) English-to-German and English-to-French training data (56 hours each, a training size similar to How2 20%). As illustrated by Table 3, MuST-C is more challenging than How2 (as confirmed by official IWSLT 2019 evaluation results (Niehues et al., 2019)), but for both language pairs, *wav2vec* significantly outperform *fbanks*. This confirms that self-supervised pre-training is useful in low resource scenarios.

## 5. Analysis of Learnt Representations

This section tries to answer the question why *wav2vec* representation performs better than *filter-bank* features in low resource settings. The following subsections present the experiments which show that *wav2vec* might be (1) better at discriminating phones, (2) better at aligning source and

*Table 3.* AST BLEU on MuST-C 56h for *EN-DE* and *EN-FR*.

| Lang | Features | tst-COMMON | tst-HE |
|------|----------|------------|--------|
| EN-DE | wav2vec | 7.56 | 7.21 |
| | wav2vec+norm | 7.83 | 8.12 |
| | fbanks | 1.50 | 1.09 |
| | fbanks+norm | 4.89 | 4.87 |
| EN-FR | wav2vec | 12.08 | 12.41 |
| | wav2vec+norm | 12.58 | 12.58 |
| | fbanks | 0.54 | 0.00 |
| | fbanks+norm | 7.10 | 6.37 |

*Table 4.* Phone error rate (PER %) on TIMIT dev and test set.

| No. | Feature | TIMIT dev | TMIT test |
|-----|---------|-----------|-----------|
| 1 | wav2vec | 13.0 | 15.0 |
| 2 | wav2vec + norm | 13.9 | 15.8 |
| 3 | fbanks | 22.2 | 24.9 |
| 4 | fbanks + norm | 20.7 | 23.5 |

target sequences, and (3) more robust to speaker variability.

### 5.1. Better phone discrimination

We first replicate an experiment from (Schneider et al., 2019) for phoneme recognition on TIMIT (Garofolo et al., 1993). Speech representations are extracted from train, dev and test split of TIMIT. A simple attentional encoder-decoder model is used: encoder with 4 BLSTM layers of hidden size 320, decoder with 1 LSTM layer and location-based attention (Luong et al., 2015). The results of Table 4 confirm that *wav2vec* representations (normalized or not) are much better at recognizing phones than *fbanks*.

### 5.2. Better source-target alignments

We evaluate the entropies of the soft alignments obtained with different speech representations in teacher forcing mode. Let $\alpha_{tj}$ be the alignment score between target token $y_t$ and source speech frame $x_j$, we evaluate the entropy

*Table 5.* Averaged entropies of soft-alignments on How2 dev and val set. AST models trained on 10% partition of How2.

| No. | Feature | How2 dev | How2 val |
|-----|---------|----------|----------|
| 1 | wav2vec | 0.66 | 0.66 |
| 2 | wav2vec + FT | 0.65 | 0.65 |
| 3 | wav2vec + norm | 0.57 | 0.57 |
| 4 | wav2vec + FT + norm | 0.51 | 0.51 |
| 5 | fbanks | 0.89 | 0.90 |
| 6 | fbanks + norm | 0.93 | 0.93 |

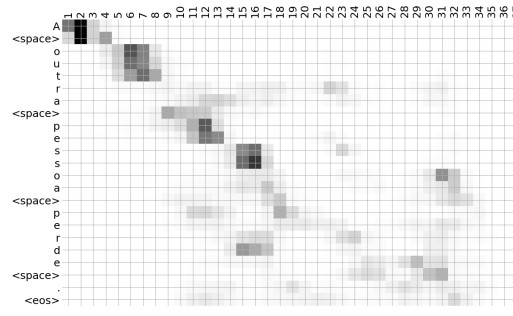

(a) wav2vec - $entropy = 0.67$

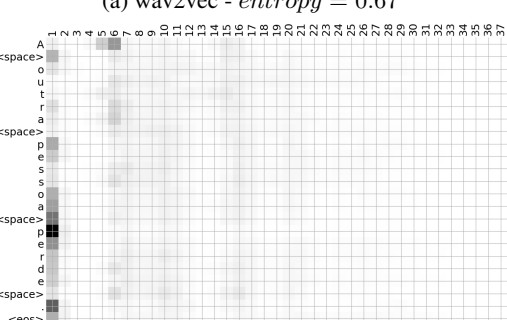

(b) fbanks - $entropy = 0.81$

*Figure 3.* Soft alignments between source speech features and target text for sentence "A outra pessoa perde."

of the probability distribution $\alpha_t$, $H_t = \sum_{j=1}^{|x|} \alpha_{tj} \log \alpha_{tj}$ for every target token. This measure is then averaged for all tokens at the corpus level (How 10%). A low entropy means the attention mechanism is confident in its source-target alignments (see example in Figure 3). Table 5 shows clearly that, in our low resource setting, *wav2vec* leads to better alignments (lower entropy) than *fbanks*. Fine-tuning and normalization of self-supervised representations also improve the soft alignments.

### 5.3. Better robustness to speaker variability

*Table 6.* Equal error rate (EER %) on the VoxCeleb1 test and LibriSpeech test sets for female (f) and male (m) speakers.

| No. | Feature | VoxCeleb | Libri (f) | Libri (m) |
|-----|---------|----------|-----------|-----------|
| 1 | wav2vec | 22.75 | 11.22 | 2.23 |
| 2 | wav2vec + norm | 20.93 | 10.54 | 1.79 |
| 3 | fbanks | 15.78 | 5.47 | 0.89 |
| 4 | fbanks + norm | 16.25 | 3.47 | 0.67 |

To investigate robustness to speaker variability, we trained several automatic speaker verification (ASV) systems using *wav2vec* or *fbanks* features. Models are trained on *LibriSpeech train-clean-360* dataset (Panayotov et al., 2015)

using Kaldi (Povey et al., 2011). ASV systems are based on x-vectors and probabilistic linear discriminant analysis (PLDA) (Snyder et al., 2018). To extract x-vectors, we used a time delay neural network (TDNN) model topology similar to the one described in (Snyder et al., 2018). Input features are *fbanks* or *wav2vec* (optionally normalized) while output corresponds to 921 speakers of the training corpus. ASV experiments are conducted on the *VoxCeleb1 test* (Nagrani et al., 2017) and *LibriSpeech test-clean* (Panayotov et al., 2015) sets.[8] ASV results (equal error rate - EER) are presented in Table 6. We observe that in all experiments, models trained on *wav2vec* features provide significantly higher EER in comparison with *fbanks*. This confirms our hypothesis that *wav2vec* representations remove speaker information from speech signal.[9]

## 6. Conclusion

We investigated the impact of self-supervised learning for end-to-end AST. It was shown that representations based on contrastive predicting coding (CPC) improve results significantly compared to baseline filter-bank, in low-medium resource conditions ($train < 100h$). Our explanation is that self-supervised representations show better phone discrimination, source-target alignments and speaker robustness.

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
