# OpenReview forum: "Investigating Self-supervised Pre-training for End-to-end Speech Translation"
_ICML.cc/2020/Workshop/SAS — SAS 2020_

### Official Review · AnonReviewer3 · 2020-06-28
**Comparing pre-trained feature extractors against filter banks**

**Rating:** 4
**Confidence:** 3

**Review:**

This paper studies self-supervised pre-training for end-to-end automatic speech translation. To do so, it compares the performance of a speech-to-text translation model when using (1) the features extracted from the self-supervised model (CPC) or (2) feature bank filters as its inputs. Their results indicate that (1) outperforms (2) in low resource settings. Further analysis indicates that this benefit might be due to pre-trained features being better at discriminating phones and at aligning source and target sequences, and being more robust to speaker variability.

The paper is written clearly and provides a clear line of argumentation towards the superiority of self-supervised pre-training for automatic speech translation in low resource settings.

The results become less clear when evaluated on the full corpus. Here, the filter bank seems to lead to slightly better results than the pre-trained features. Additionally, the paper's claim to achieve near state of the art performance seems to fall out of the context of the rest of the paper. Firstly, self-supervised pre-training seems to perform best (comparatively) in the low resource setting. But instead of comparing against other methods in this setting, the paper tries to achieve SOTA on the full corpus. Second, while pre-trained features are the method proclaimed superior throughout the paper, here an ensemble of all models (including the feature bank filters) is used. This seems to indicate that the two types of features provide complementary inputs that improve performance when combined. Unfortunately, this is not investigated further. Along this line, it remains unclear whether the analysis of the learned representations (section 5) has been conducted in the low or high resource setting.

The treatment of training/validation/development/testing splits of the datasets is confusing to me. Instead of using the provided validation set of How2, the paper claims to use its own development set and evaluates results on the validation set. Additionally, results are provided for both dev and test set in table 4 and for dev and val set in table 5.

Finally, the paper exceeds the five page limit.

---

### Official Review · AnonReviewer2 · 2020-06-29
**Good paper and detailed analysis on the use of CPC model in end2end speech translation**

**Rating:** 8
**Confidence:** 5

**Review:**

This paper investigates the application of self-supervised pre-training to end-to-end speech translation. In particular, it presents an application of the contrastive predictive coding (CPC) model that gives the possibility of leveraging massive audio data without the need of the textual transcriptions. An extensive set of experiments and analysis is reported showing that i) the CPC model is particularly beneficial when limited quantities of training data is available and ii) the use of wav2Vec allows the speech translation model to have a better representation of the input audio compared to filter banks.

The paper is clear and well-written. The addressed topic is interesting for the ST community because the paucity of speech-translation training data calls for different approaches to use alternative sources of information (e.g. only audio data). The motivation and the experimental parts are robust and appropriate. The analysis of the learnt representation tries to shed light on the improvements in performance obtained in low-resource settings.

The main limitation of this paper could be the novelty because the paper extends the work by Chung & Glass (2019 and 2020). However, the use of different training data sizes and the investigation of the learnt representation make the paper interesting and quite distinguishable from the previous literature.

Comments and questions:

-) Why did the authors choose CPC model instead of the autoregressive predictive coding (ACPC)? Is it only a matter of simplicity of the former? or there is evidence that the former outperforms the latter? The motivation should be mentioned in the paper.

-) It is not clear if the (Inaguma et al., 2020) models resulting in the best BLEU score on How2 are trained only using How2 or with more speech-translation data. This should be specified at the end of section 4.3.

-) The best systems at the IWSLT workshop last year used much more ST data and different data augmentation techniques such as knowledge distillation. The ensemble of wav2vec and fbanks approaches the best performing system on the How2 dataset. Although it is clear that the CPC model has fewer requirements in terms of task-specific data, it would be interesting and useful for the community to have a proper comparison/combination of the CPC model with these methods both in low and high-resource scenarios

---

### Official Review · AnonReviewer4 · 2020-07-01
**Promising approach for speech translation, but the experimental setting can be improved**

**Confidence:** 4
**Rating:** 5

**Review:**

This paper compares two audio representations for automatic speech translation, where the models considered translate directly from the audio source. The two representations are 1. audio features obtained by Mel filter-banks, and 2. *context* features obtained via a network trained with contrastive predicting coding (CPC). The goal is to understand whether self-supervised training can lead to better representations than the widely used Mel filter-banks for the task at hand. Additionally, self-supervised learning can also replace the pre-training step performed on ASR, which is currently considered beneficial for the translation task. The advantage of self-supervised learning would be the possibility of pre-training a network using only unlabeled data.
However, the results are favorable for the self-supervised-learning approach only in the **simulated** low-resource scenario and under the assumption that transcripts are not available for the audio used for training. The learning curves plotted in Figure 2 show a faster convergence for wav2vec, but this advantage vanishes as the size of the training set increases. Under these assumptions, I cannot figure out why the authors decided, after the experiments on the How2 corpus, which is one of the largest available for the task, to confirm the results on a simulated low-resource setting using MuST-C. I would rather have expected experiments on a really low-resource language pair, which can be easily found among the corpora for speech translation.
Moreover, it is worth noticing that, in the high-resource settings showed in Table 2, the results of all the models are very similar although the models using wav2vec had access to more data than their counterparts.
For what concerns the analysis presented in Section 5, I have contrasting ideas. I think that it is useful to see the potentiality of the two audio representations to get more insights about them. However, the two methods under analysis are not new and the cited literature presented the advantages of CPC and similar methods. Nonetheless, such analysis could still be useful if framed within the speech translation task. In contrast, in Sections 5.1 and 5.3 new models are trained for different tasks, showing the advantages of the wav2vec representation that can lead to better results in the low-resource setting. However, the models used for these experiments are quite different from the ones used for speech translation. Then, although the results are significant, in particular for what concerns sensitivity to speaker variability, it is not clear whether the advantage for speech translation can be explained in this way. On the other hand, I do not think that the results presented in Section 5.2 are really useful, as they compare the attention distribution of models of significantly different quality. I would not expect a reliable attention from a model that scores 1-2 BLEU points.

Overall:
+ The paper is clearly written and the study is well-described.
+ It presents an attempt to use self-supervised learning to replace pre-training on the ASR task with a task that does not need labeled data.
+ The discussed approach is not new, even in this task, but this paper tries to go deeper than the previous work for the task at hand.
+ The significance of this work is undermined by the negative results for settings with enough data. I think that self-supervised pre-training can lead to positive results also in those settings, as speech translation is generally a low-resourced task.
+ The focus on the very low-resource setting is weakened by the experiments that only simulate that setting, while real corpora are available, and the absence of a comparison with methods for such settings, as the cross-lingual pre-training presented by Bansal et al. in  Bansal, Sameer, et al. "Pre-training on high-resource speech recognition improves low-resource speech-to-text translation." arXiv preprint arXiv:1809.01431 (2018).
+ The analysis can be improved, for example by inspecting the speech translation models for the other task, as it can confirm the results found by training additional models.

---

### Official Review · AnonReviewer5 · 2020-07-01
**Good, with some major drawbacks**

**Confidence:** 4
**Rating:** 6

**Review:**

This paper evaluates the impact of the self-supervised pre-training for the task of speech translation (speech -> text in different language). First, the paper extracts wav2vec features from a large Librispeech corpus. Then these features are compared to fbanks on a smaller speech translation corpus. The authors observe that fbanks perform worse than wav2vec in the cases when the dataset is small.

In general, I think this is an informative paper. The experimental evaluation deserves the attention of the community.

However, I found the analysis section answering a wrong question. The question the paper states first is why wav2vec work well for small datasets. On the contrast, the analysis section experiments are not specific to small datasets. Also, 5.2 and 5.3 might confuse the cause and effect. Is it possible that the alignments are bad because the model performs bad, not other way around?

All in all, I find the paper good enough for the workshop. There are some major drawbacks which I recommend to consider and perhaps, conduct more experiments.

---

### Decision · Program_Chairs · 2020-07-01

**Decision:**

Accept

**Comment:**

Dear author(s),

Thank you very much for your submission at the ICML2020@SaS workshop (https://icml-sas.gitlab.io/). Based on the scores assigned by the reviewers, we are happy to notify you that your paper was accepted for the workshop.

It turned out that this paper has mixed reviews, but we think it is highly relevant for the workshop.

Please, address the comments of the reviewers and submit the camera-ready version by July 8. We ask the authors to record a 15min video for your talk. At the workshop, we will have the pre-recorded video as well as a live QA session. It is important to keep this time limit, otherwise, your talk will be automatically cut. The deadline for uploading the video is July 8. The detailed instructions for uploading will follow.

Feel free to contact us for any questions!

Best,

The ICML20@SaS organizers:
Mirco Ravanelli
Titouan Parcollet
Dmitriy Serdyuk
Devon Hjelm
Bhuvana Ramabhadran